# Neurotrophin Analog ENT-A044 Activates the p75 Neurotrophin Receptor, Regulating Neuronal Survival in a Cell Context-Dependent Manner

**DOI:** 10.3390/ijms241411683

**Published:** 2023-07-20

**Authors:** Maria Anna Papadopoulou, Thanasis Rogdakis, Despoina Charou, Maria Peteinareli, Katerina Ntarntani, Achille Gravanis, Konstantina Chanoumidou, Ioannis Charalampopoulos

**Affiliations:** 1Department of Pharmacology, Medical School, University of Crete, 71003 Heraklion, Greece; medp2011866@med.uoc.gr (M.A.P.); trogdak@gmail.com (T.R.); dcharou@gmail.com (D.C.); med5p1060194@med.uoc.gr (M.P.); bio3095@edu.biology.uoc.gr (K.N.); gravania@uoc.gr (A.G.); k.chanoumidou@gmail.com (K.C.); 2Institute of Molecular Biology & Biotechnology (IMBB), Foundation for Research and Technology-Hellas (FORTH), 70013 Heraklion, Greece

**Keywords:** neurotrophins, p75 receptor, steroidal synthetic analogs, TrkB receptor, cell death, human-induced Pluripotent Stem Cells, neural stem cells

## Abstract

Neuronal cell fate is predominantly controlled based on the effects of growth factors, such as neurotrophins, and the activation of a variety of signaling pathways acting through neurotrophin receptors, namely Trk and p75 (p75NTR). Despite their beneficial effects on brain function, their therapeutic use is compromised due to their polypeptidic nature and blood–brain-barrier impermeability. To overcome these limitations, our previous studies have proven that DHEA-derived synthetic analogs can act like neurotrophins, as they lack endocrine side effects. The present study focuses on the biological characterization of a newly synthesized analog, ENT-A044, and its role in inducing cell-specific functions of p75NTR. We show that ENT-A044 can induce cell death and phosphorylation of JNK protein by activating p75NTR. Additionally, ENT-A044 can induce the phosphorylation of TrkB receptor, indicating that our molecule can activate both neurotrophin receptors, enabling the protection of neuronal populations that express both receptors. Furthermore, the present study demonstrates, for the first time, the expression of p75NTR in human-induced Pluripotent Stem Cells-derived Neural Progenitor Cells (hiPSC-derived NPCs) and receptor-dependent cell death induced via ENT-A044 treatment. In conclusion, ENT-A044 is proposed as a lead molecule for the development of novel pharmacological agents, providing new therapeutic approaches and research tools, by controlling p75NTR actions.

## 1. Introduction

To a great extent, the fate of neuronal cells is controlled based on the effects of growth factors, such as neurotrophins. The neurotrophin family consists of the secreted growth factors BDNF, NGF and NT3/4, whose main functions are to induce neuronal survival and regeneration, thus allowing neurons to survive, differentiate and grow [1,2]. All of these biological effects are specifically mediated through binding of neurotrophins to the high-affinity Trk receptors and the low-affinity p75 pan-neurotrophin receptor. These receptors can function independently or interact naturally and functionally with each other, doing so in ways that can change the signaling characteristics of each receptor [3].

The pan-neurotrophin receptor p75NTR is a member of the TNF death receptors superfamily. Its wide expression in many cell types and its controversial signaling pathways, have increased scientific interest in this receptor. While the most distinguished effects of p75NTR are related to programmed cell death, recent studies have revealed its role in regulating cellular responses, such as cellular proliferation and survival [4,5,6], particularly when acting as a co-receptor of Trk receptors for mature neurotrophins [3,7]. The p75NTR signaling properties are predominantly mediated via the death domain of the receptor, which functions based on the recruitment or release of specific intracellular interactors in its surface [8,9].

Neutrophic effects on brain function and repair have highlighted these growth factors and their receptors as promising therapeutic candidates for treatment of neurodegenerative diseases. However, despite their proven beneficial effects on the survival and protection of neurons, their therapeutic usefulness is limited due to their polypeptide nature and limited BBB permeability. Attempts to bypass these limitations, such as ICV infusion [10,11] or intranasal delivery of BDNF and NGF [12,13], have shown limited therapeutic effects in clinical trials, as only a small amount of the applied drug finally reaches the CNS via intranasal delivery.

Our laboratory’s previous research efforts revealed the ability of the endogenous neurosteroid dehydroepiandrosterone (DHEA) to block neuronal apoptosis by binding and activating the NGF receptors, namely TrkA and p75NTR [14,15,16]. However, DHEA is metabolized in vivo using androgens and estrogens, and it can, thus, increase the risk of hormone-dependent cancer development. To overcome these problems, small molecules—chemical analogs of DHEA—that sustain the neurotrophin effects but lack the undesired endocrine effects could be pharmacologically useful. BNN27, which is a C-17-spiro derivative of DHEA, was one of the first chemical compounds to be tested, and it has been shown to act through the selective activation of both p75NTR and TrkA receptors [17,18]. More recently, a library of novel small molecules based on the aforementioned molecules were synthesized, which demonstrated improved pharmacological properties and lacked the endocrine side effects of the parental molecule.

In the present study, we focus on the biological evaluation of the compound ENT-A044, which is a chemical entity derived from DHEA, with a C17 structural modification. We investigated its interaction with the neurotrophin receptors, especially p75NTR and TrkB. In this process, ENT-A044 stood out due to its effects on p75NTR, among several others DHEA synthetic analogs, which were screened for their ability to activate neurotrophin receptors. In this paper, we report its role in activating death signaling pathways in p75NTR-transfected HEK293T cells challenged with serum deprivation, as well as survival pathways in P7 mouse hippocampal neural stem cells that express the TrkB receptor. Additionally, we identify the activated signaling pathways mediated by the p75 receptor’s interactor protein RIP2. Notably, we report, for the first time, the expression of p75NTR in human-induced Pluripotent Stem Cells (iPSCs)-derived Neural Precursor Cells (NPCs), as well as the subsequent death signaling pathways mediated via treatment with ENT-A044. Our results propose that ENT-A044 can not only be used as a lead molecule for the development of non-selective agonists that act on p75NTR and leading on cell death, but could also be exploited as a research tool to decipher the pleiotropic signaling pathways mediated by p75NTR in a cell-specific manner.

## 2. Results

### 2.1. ENT-A044 Induces Cell Death of PC12 Cells

PC12 cells, which form a cell line that expresses TrkA and p75NTR [19], are commonly used to study neurotrophin-dependent effects by evaluating their neuroprotective actions. We tested ENT-A044 in PC12 cells under serum starvation conditions in order to induce cell death. Cells were treated for 24 h with NGF or the compound in the absence of serum, and the CellTox cytotoxicity assay was performed to identify apoptotic cells. ENT-A044 did not protect PC12 cells from cell death caused by serum deprivation, and, surprisingly, it significantly augmented cell death (Figure 1a). Given that our compound promotes cell death in PC12 cells that express the TrkA receptor and p75NTR, we assumed that the observed cell death was probably mediated by the actions of the pro-apoptotic p75 death receptor. To investigate the underlying mechanism, we first explored the possibility that ENT-A044 interacts with the TrkA neurotrophin receptor and its downstream signaling kinases Akt and Erk1/2. Thus, PC12 cells were starved of serum for 4 h and then treated with NGF (100 ng/mL) or ENT-A044 (500 nM) for 20 min. In parallel, other analogs of the same chemical library were also tested. Western blot analysis revealed that ENT-A044 cannot induce the phosphorylation of kinases Akt and Erk1/2, which are known to be activated upon TrkA activation and have been associated with NGF promotion of cell survival (Figure 1b), while other compounds of similar structure were able to activate these kinases [20]. Thus, considering the inability of ENT-A044 to activate the TrkA receptor, the observed cell death in PC12 is probably mediated by p75NTR.

### 2.2. ENT-A044 Activates TrkB Receptor and Its Downstream Signaling Kinase Akt and Protects NIH-3T3-TrkB Cells from Serum Deprivation-Induced Cell Death

Next, we investigated the capacity of ENT-A044 to activate the TrkB receptor, before emphasizing the effects of ENT-A044 on p75NTR. For this purpose, we used NIH-3T3 cells that were stably transfected with the TrkB receptor—NIH-3T3-TrkB—and treated for 24 h with BDNF or ENT-A044 in the absence of serum. Cells were then stained with CellTox reagent and Hoechst to identify apoptotic cells. ENT-A044 seems to protect NIH-3T3-TrkB cells from induced cell death. In order to confirm the involvement of TrkB in the protective effect of ENT-A044, we treated cells with either BDNF or the compound in the presence of one selective TrkB inhibitor, ANA-12. As it is depicted on the right panels of Figure 2a,b, the inhibition of TrkB receptor totally abolishes the neuroprotective effect of ENT-A044 and BDNF, clearly indicating that TrkB receptor is the exclusive mediator of cell protection. To further evaluate this hypothesis, we tested whether ENT-A044 activates the TrkB downstream signaling kinase Akt. For this reason, we used primary astrocytes isolated from the cortex of post-natal day 2 (P2) mouse pups, which endogenously express TrkB. Cells where starved of serum for 4 h, and treatment with BDNF (500 ng/mL) or ENT-A044 (500 nM) occurred for 20 min. Western blot analysis revealed that ENT-A044 induces the phosphorylation of TrkB receptor and the kinase Akt (Figure 2c,d), suggesting that ENT-A044 can activate TrkB receptor and its downstream signaling.

### 2.3. ENT-A044 Promotes Cell Death by Activating p75NTR in Transiently Transfected HEK293T Cells

Following the observed cell death in PC12 cells and TrkB activation by ENT-A044, we proceeded to study our compound’s implementation in p75NTR activation. HEK293T cells were transiently transfected with the plasmid for the expression of human p75NTR. Treatment of these cells with ENT-A044 revealed an increase in observed death following serum deprivation-induced cell death (Figure 3a,b). HEK293T cells that were not transfected with p75NTR plasmid (wild type HEK293T cells) were also treated with ENT-A044, having no differences from the serum-free condition (Appendix A). Phosphorylation of JNK protein was also measured in order to define the exact signaling pathways that are involved in the induction of cell death. JNK, which is a kinase of the mitogen-activated protein kinases, is activated upon p75NTR activation, while phosphorylates cJun and triggers apoptotic signaling pathways via up-regulation of proapoptotic genes [21,22]. After 30 min of treatment with NGF or our compound, phosphorylation of JNK was increased (Figure 3c). Thus, ENT-A044 can activate JNK-mediated death signaling pathways via p75NTR.

To better characterize the p75NTR signaling pathways that are activated by ENT-A044, we performed an immunoprecipitation assay. p75NTR lacks intracellular enzymatic activity and its signaling is dependent on interactions between proteins, like TRAF6 and RIP2, and the receptor’s death domain. By performing co-immunoprecipitation experiments for p75NTR and its interactors TRAF6 and RIP2, we observed that there was no significant recruitment of TRAF6 upon p75 receptor activation using our compound (Figure 3d), while the RIP2 protein exhibits a significant interaction with the p75NTR in treated with ENT-A044 samples (Figure 3e). Thus, we clearly show that ENT-A044 can induce multiple, though not all, signaling cascades of the p75NTR, offering a valuable experimental tool for deciphering signaling properties of the receptor.

As HEK293T cells only express p75NTR, it seems that ENT-A044 can induce cell death when only p75NTR is being expressed, while it has opposite effects when the TrkB receptor is also co-expressed, as in the case of astrocytes. Due to those results, we examined the effects of ENT-A044 in a system in which both TrkB and p75NTR are expressed. Firstly, we used co-transfected HEK293T cells with both TrkB and p75NTR plasmids. Cell tox assay then showed cell survival or death based on which receptor was present. More specifically, when both receptors were present, ENT-A044 predominantly led to survival (Appendix A). This result is a clear indication that ENT-A044-dependent signaling of the TrkB receptor can overcome the death signaling pathway that is triggered by p75NTR in a cell-specific manner.

### 2.4. ENT-A044 Promotes Cell Survival in P7 Mouse Hippocampal NSCs

Furthermore, we used primary cultures from P7 mouse hippocampal NSCs that express both TrkB and p75NTR (Appendix A), showing that, when both of the receptors are present, ENT-A044 treatment for 48 h can lead to survival signaling (Figure 4). P7 hippocampal NSCs were also tested upon compound treatment for 24 h, having no significant results (although there is a trend of protection), which was probably due to the fact that significant cell death was not observed before 48 h (Appendix A).

### 2.5. ENT-A044 Shows a Strong Pro-Apoptotic Effect on Human NPCs

Lastly, we focused on human cells of neural origin and tested the effect of ENT-A044 on Human Neural Progenitor cells (NPCs). NPCs were generated via human-induced Pluripotent Stem Cells (hiPSCs) and express both TrkB and p75NTR receptors (Appendix A). NPCs treatment with ENT-A044 for 48 h had a detrimental effect on cell survival, as revealed via cell tox cytotoxicity assay (Figure 5a,b). The apoptotic effect of ENT-A044 is in line with the increased levels of pJNK found in NPCs upon treatment with the compound for 24 h. Increased pJNK was also observed upon cells’ treatment with NGF, suggesting the activation of an apoptotic signaling pathway totally mediated by p75NTR in these cells (Figure 5d). On the other hand, we observed that treatment with BDNF for 48 h could not induce cell death mediated by p75NTR, which occurs independently of its expression in human NPCs (Figure 5a,c). Thus, ENT-A044 leads to cell death mediated by p75NTR in human NPCs, although these cells express both p75NTR and the TrkB receptor, indicating significant differences between human and mouse neuronal cell functions that are mediated by neurotrophin receptors.

## 3. Discussion

Our research group has shown that the endogenous neurosteroid dehydroepiandrosterone (DHEA), which consists of a molecule of small size with a lipophilic structure, can act on TrkA and p75NTR and prevent neuronal apoptosis [14,16]. However, DHEA is metabolized in vivo to estrogens and androgens, affecting the endocrine system; therefore, the long-term use of DHEA as a potential treatment for neurodegeneration is problematic, particularly in patients with genetic predisposition to hormone-dependent tumors (breast, endometrium, ovaries, prostate, etc.) [23,24]. To overcome these limitations of DHEA and take advantage of the beneficiary effects of this small-sized molecule, Calogeropoulou et al. developed a chemical library of synthetic C17-derivatives of DHEA [24] with anti-apoptotic properties, which lack androgenic/estrogenic actions. These novel molecules are BBB-permeable, and, most importantly, they act as neurotrophin mimetics to address the neuroprotective and neurogenic effects of endogenous neurotrophins. BNN27, which is a C17-spiroepoxy steroid derivative, was shown to specifically interact with and activate the TrkA receptor that induces phosphorylation of TrkA tyrosine residues and down-stream neuronal survival-related kinase signaling [18]. It can also interact with p75NTR, inducing the recruitment of effector proteins RIP2 and TRAF6 to the receptor and the release of RhoGDI in primary granular cells of the cerebellum, as well as inducing neuroprotective effects [17]. Recently, Rogdakis et al. showed that ENT-A013, which is a chemically stable and more potent compound than BNN27, selectively activates TrkA receptor and exerts neuroprotective and anti-amyloid actions [20].

The present study provides evidence of one of these synthetic compounds. Although it activates both TrkB and p75NTR neurotrophin receptors, it exerts differential signaling and cellular effects in a cell-specific manner, depending on which receptor is expressed and, surprisingly, the species origin—mouse or human—of the neuronal cell. Notably, ENT-A044 is a compound that is structurally similar to ENT-A013, which is a well characterized TrkA activator derived from the same synthetic procedures [20]. However, ENT-A044 cannot activate TrkA, as is obvious from the total absence of phosphorylation of Akt and Erk kinases, which are the most significant downstream signals of the TrkA receptor in TrkA-expressing PC12 cells. On the other hand, we observed TrkB phosphorylation and Akt phosphorylation in TrkB-expressing cells, showing that signaling pathways mediated by TrkB are activated upon ENT-A044 treatment. Exploring in depth our initial observation that ENT-A044 induces cell death on PC12 cells (which are expressing both TrkA and p75NTR) even after serum deprivation-induced apoptosis, we investigated whether this effect was particularly mediated by p75NTR. In agreement with this hypothesis, in transiently transfected HEK293T cells that only express p75NTR, our compound increased the levels of cell death, while in naive HEK293T cells, ENT-A044 had no effect on cell death. Thus, cell death that results from treatment with ENT-A044 is clearly driven by p75NTR activation. More specifically, this death signal is primarily executed through the activation of JNK protein, which is known to be implicated in cell death signaling pathways that originate from p75NTR [9].

p75NTR, as a member of TNFR superfamily, includes a Death Domain at its intracellular part, which affords most of its signaling functions [9]. Depending on the ligand that activates the receptor and the cell type, its actions range from cell survival to cell death, while the co-expression of Trk receptors or other co-receptors, like sortilin or Nogo, can also differentiate the final cellular outcome [5,25]. Enhancement of neuronal survival by p75NTR has been associated with the activation of the transcription factor NFkB [26] after RIP2 recruitment. More specifically, activation of the transcription factor NFkappaB upon the recruitment of RIP2 protein to p75NTR leads to survival signals on specific neuronal populations [27,28]. ENT-A044 efficiently induced the association and interaction between p75NTR and its effector protein RIP2, as well as in transiently transfected HEK293T-p75 cells, and had no significant effects on interaction with TRAF6 protein, thus inducing cell apoptosis. Since ENT-A044 activates both TrkB and p75NTR, resulting in cell survival, an important question has immerged regarding the potential triangle interaction between RIP2, p75NTR and the TrkB receptor, which combinatorically leads towards survival signaling pathways. In transiently transfected HEK293T-TrkB/p75 cells and primary cultures of mouse P7 hippocampal NSCs, ENT-A044 can trigger the survival signaling pathway activated by TrkB receptor and overcome cell death, which is activated by p75NTR in the absence of TrkB. In the last case, p75NTR-dependent induction of cell death overcomes the protective RIP2 signaling, which most probably occurs due to significant induction of the pro-apoptotic JNK signal. Figure 6 summarizes all signaling pathways of ENT-A044 and its cellular effects.

Finally, for the first time based on our knowledge, we showed that p75NTR is expressed in human NPCs generated by hiPSCs. These neural precursors also express the TrkB receptor. In contrast to our previous results in mouse cell or cell lines, where both receptors were co-expressed, ENT-044 has a totally different effect on human neural stem cells, leading to significant induction of cell death. Given that treatment with NGF also increased the levels of pJNK, this apoptotic role is exclusively mediated by p75NTR, since TrkA receptor is not expressed. These different cellular responses between mouse and human neural cells could reflect differences between endogenous neurotrophins and ENT-A044 in terms of their affinity with the responsive receptors, as well as different signaling capabilities due to the diverse intracellular protein cargo (Figure 6). In addition, based on previous studies, it could be possible that ENT-A044 is modulating the receptor’s structure, as well as the interaction between p75NTR and its co-receptors, like Trks or sortilin [29]. In support of this hypothesis, we have shown that our initially tested compound—BNN27—has different binding sites in TrkA and p75NTR than NGF [18].

Last but not least, taking advantage of ENT-A044 mediating cell death through p75NTR, we could therapeutically exploit this compound against diseases like cancer, targeting the elimination of cancer cells. p75NTR’s role in cancer is controversial. On one hand, it has been shown to act as a tumor suppressor and a prognostic factor in cancers, while, on the other hand, it participates in tumor aggressiveness [30,31,32,33,34,35] Goh et al. showed that NSC49652, which is a small molecule that targets the p75NTR transmembrane domain, can induce apoptosis of melanoma cells acting through p75NTR, taking advantage of the fact that several cancers, such as neural crest-derived melanoma, express high levels of p75NTR [36].

All aforementioned observations highlight the controversial nature of p75NTR, and further analysis on the signaling pathways and mechanisms that are being activated by this receptor are necessary. One limitation of p75NTR is its pleiotropic signaling cascades and the multi-variant cellular responses that occur upon its activation. Thus, it would be really interesting in our future studies to include data from -omics analysis at a single cell level in the presence of endogenous ligands and synthetic agonists. Additionally, in vivo studies of disease mouse models related to the receptor’s effects, like Alzheimer’s Disease, could offer more precise documentation regarding the receptor’s role in neurodegeneration. Molecules like ENT-A044 can offer a valuable experimental tool that dissects these receptor’s pleiotropic functions between different cell types and species. Additionally, small-sized compounds with desirable pharmacological properties, like ENT-A044, could offer novel therapeutic approaches against neurodegenerative diseases or tumor and cancer growth.

## 4. Materials and Methods

### 4.1. Cell Lines

HEK293T cells were obtained from LGC Promochem GmbH (Teddington, UK) and cultured using DMEM medium (11965084, Gibco, Grand Island, NY, USA) that contained 10% Fetal Bovine Serum (10270106, Gibco, Grand Island, NY, USA) and 100 units/mL Penicillin and 0.1 mg/mL Streptomycin (15140122, Gibco, Grand Island, NY, USA) at 5% CO_2_ and 37 °C. They were transiently transfected with human p75NTR and TrkB plasmids, as well as TRAF6 and RIP2 plasmids, using Turbofect Transfection Reagent (R0531, Thermo Fisher Scientific, Waltham, MA, USA) based on manufacturers’ instructions. Plasmids that expressed p75NTR and TrkB were previously described in [14,17,18]. Transfected cells were typically used on the second day following transfection. NIH3T3 cells were grown in the same medium and stably transfected with human TrkB plasmid. PC12 cells were cultured under the same medium, which contained 10% Horse Serum and 5% Fetal Bovine Serum. All cells were used in the passages 5–20.

### 4.2. Primary Cell Cultures

Primary hippocampal NSCs were isolated from post-natal day 7 (P7) mouse pups (C57BL/6J, The Jackson Laboratory, Bar Harbor, ME, USA). These cells were grown in DMEM/F12 medium that contained B27 supplement without vitamin A, D-glucose, Primocin (0.1 mgr/mL), FGF (0.02 μgr/mL), EGF (0.02 μgr/mL) and Heparin (0.1 mgr/mL) at 5% CO_2_ and 37 °C. They were checked every day for neurosphere formation and their morphology—primary neurospheres observed after 5–7 days—when relatively large but bright neurospheres have formed, and a passage of the cells was necessary, which occurred using accutase. Glial cultures were isolated from the cortex of post-natal day 2 (P2) mouse pups (C57BL/6J, The Jackson Laboratory, Bar Harbor, ME, USA). Cells were grown in high glucose DMEM medium that contained 200 U/mL penicillin, 200 μgr/mL streptomycin and 10% fetal bovine serum (FBS). At day 7, anti-mitotic agent Ara-C was added to the medium at a final concentration of 10 μM and maintained for 3 to 4 days to target the highly proliferative microglial cells. When Ara-C was removed, primary astrocytes reached a purity of 97%, and they were cultured using 5% CO_2_ and 37 °C. For all experiments, cells were plated on PDL/laminin, and the assays were performed after 24 h.

### 4.3. Generation and Culture of Human Neural Progenitor Cells (NPCs)

NPCs were generated from human-induced Pluripotent Stem Cells (iPSCs), as previously described in [37]. Briefly, iPSCs colonies were sectioned and enzymatically detached from mouse embryonic fibroblasts. The pieces of iPSCs colonies were collected and cultured as embryoid bodies, in suspension, in a medium that consisted of knockout DMEM (Invitrogen, Waltham, MA, USA), 20% (*v*/*v*) Knockout Serum Replacement (Invitrogen), 1 mM of β-mercaptoethanol (Invitrogen), 1% non-essential amino acids (NEAA; Invitrogen), 1% penicillin/streptomycin/glutamine (PAA) supplemented with 10 µM of SB-431542 (Ascent Scientific, Bristol, UK), 1 µM of dorsomorphin (Tocris, Bristol, UK), 3 µM of CHIR99021 (CHIR; Axon Medchem, Reston, VA, USA) and 0.5 of µM purmorphamine (Alexis, Miami, FL, USA). After two days, the medium was changed to N2B27 medium, which contained ½ DMEM-F12 (Invitrogen) and ½ Neurobasal (Invitrogen) supplemented with N2 supplement (Invitrogen), B27 supplement lacking vitamin A (Invitrogen) and 1% penicillin/streptomycin/glutamine supplemented with the aforementioned small-molecules. On day 4, SB-431542 and dorsomorphin were replaced by 150 µM of ascorbic acid (AA; Sigma, St. Louis, MO, USA). On day 6, the spheres were cut into smaller pieces and plated on Matrigel-coated plates (BD Biosciences, England, UK) in the N2B27 medium supplemented with 3 µM of CHIR, 0.5 µM of SAG (Cayman Chemical, Ann Arbor, MI, USA) and 150 of µM AA. When confluent, cells were split via treatment with accutase (Sigma, St. Louis, MO, USA). The identity of the cells was verified via immunocytochemistry for NESTIN and SOX1.

### 4.4. Treatment with Compound ENTA-044

Cells were starved of serum for four hours so that they could be synchronized prior to treatment and then stimulated with 100 ng/mL of NGF or/and 500 ng/mL of BDNF and the examined compound ENT-A044 (500 nM). The activation of the tested receptors and the Immunoprecipitation assay treatment lasted for 20 min. For the phosphorylation of JNK protein, treatment lasted for either 30 min or 24 h, as indicated in [9]. NPCs were treated with 1 μM of ENT-A044 for 48 h via the cell tox assay and 24 h via Western blot analysis.

### 4.5. Immunoprecipitation and Immunoblotting

Cells were suspended in Pierce™ IP Lysis Buffer (87788, Thermo Fischer Scientific, Waltham, MA, USA) supplemented with protease inhibitors (539138, Calbiochem, Darmstadt, Germany) and phosphatase inhibitors (524629, Calbiochem, Darmstadt, Germany). Lysates were pre-cleared for 1 h with protein G-plus Agarose beads (sc-2002, Santa Cruz Biotechnology, Inc., Dallas, TX, USA) and immunoprecipitated with p75NTR antibody (1:100, ab6172, Abcam plc., Cambridge, UK) overnight at 4 °C. Protein G-plus agarose beads were incubated with the lysates for 4 h at 4 °C via gentle shaking. Beads were collected via centrifugation, re-suspended in 2× SDS loading buffer and subjected to Western blot analysis against Traf6 antibody (1:2000, ab33915, Abcam, plc., Cambridge, UK) and RIP2 (1:1000, ADI- AAP-460, Enzo Life Sciences Farmingdale). For immunoblot (IB) analysis, the beads were suspended in sodium dodecyl sulfate-loading buffer and separated through SDS-PAGE. Proteins were transferred to nitrocellulose membranes and blotted with the corresponding antibodies. Whole cell lysates were subjected to Western blot analysis against phosphorylated TrkB (1:1000, ABN1381, Sigma-Aldrich, St. Louis, MO, USA), TrkB (1:1000, 07-225-I, Sigma-Aldrich, St. Louis, MO, USA), p75NTR (1:1000, 839701, Biolegend, Inc., San Diego, CA, USA), Traf6 (1:2000, ab33915, Abcam, PLC., Cambridge, UK), phosphorylated Akt (1:1000, 9721S, Cell Signaling Technology, Danvers, MA, USA), phosphorylated Erk1/2 (1:1000, 9101S, Cell Signaling Technology, Danvers, MA, USA), total Akt (1:1000, 4691S Cell Signaling Technology, Danvers, MA, USA), total Erk1/2 (1:1000, 4695S, Cell Signaling Technology, Danvers, MA, USA), Actin (1:2000, sc-47778 Santa Cruz Biotechnology, Inc., Dallas, TX, USA), pJNK (1:1000, 4668 Cell Signaling Technology, Danvers, MA, USA) and tJNK (1:1000, 9252 Cell Signaling Technology, Danvers, MA, USA). Immunoblots were developed using the ECL Western Blotting Kit (Thermo Fisher Scientific), and Image analysis and quantification of band intensities were performed with ImageJ Software. For the phosphorylated forms of proteins, such as Akt, Erk1/2, TrkB and JNK, the analysis was derived from the phosphorylated fraction relative to the total fraction. For the immunoprecipitation assay (activation of p75NTR according to TRAF6 and RIP2 interactions), the analysis was derived from the immunoprecipitated fraction relative to the total fraction.

### 4.6. Cell Tox Assay

After 24 h of treatments, we used the CellTox™ Green Cytotoxicity Assay kit (G8742, Promega Corporation, Maddison, WI, USA) to assess the survival of NIH-3T3, PC12, HEK293T and P7 hippocampal NSCs under conditions of serum and EGF/FGF deprivation, respectively. NPCs were studied after 48 h. Cells were plated in 96-well plates, starved for 4 h and treated with NGF (100 ng/mL) and/or BDNF (500 ng/mL) and compound ENT-A044(500 nM) in the presence or absence of p75NTR inhibitor MC-192 (2.5 ng/mL, ab6172, Abcam, plc., Cambridge, UK) and TrkB inhibitor 100 μM of ANA-12 (SML0209, Sigma-Aldrich, Burlington, MA, US) for 24 h. CellTox assay reagents and Hoescht (1:10,000, H3570, Invitrogen, Waltham, MA, USA) were added to each well for 30 min, and cells were imaged using a fluorescent microscope (Zeiss AXIO Vert A1, Zeiss, Jena, Germany). Positive cells for cell tox reagent were normalized to reflect the total number of cells per image. We also refreshed the examined compound, as well as BDNF or/and NGF and HEK293T cells and P7 hippocampal NSCs, and they were imaged again after 48 h.

### 4.7. Statistical Analysis

All values are expressed as the mean ± SEM. Student’s *t*-test was used for the comparison of two groups, and one-way ANOVA was used for multiple group comparisons. A *p* < 0.05 was considered to mark statistical significance. Statistical analysis was performed using GraphPad Prism 7 (GraphPad Software Inc., San Diego, CA, USA).

## Figures and Tables

**Figure 1 ijms-24-11683-f001:**
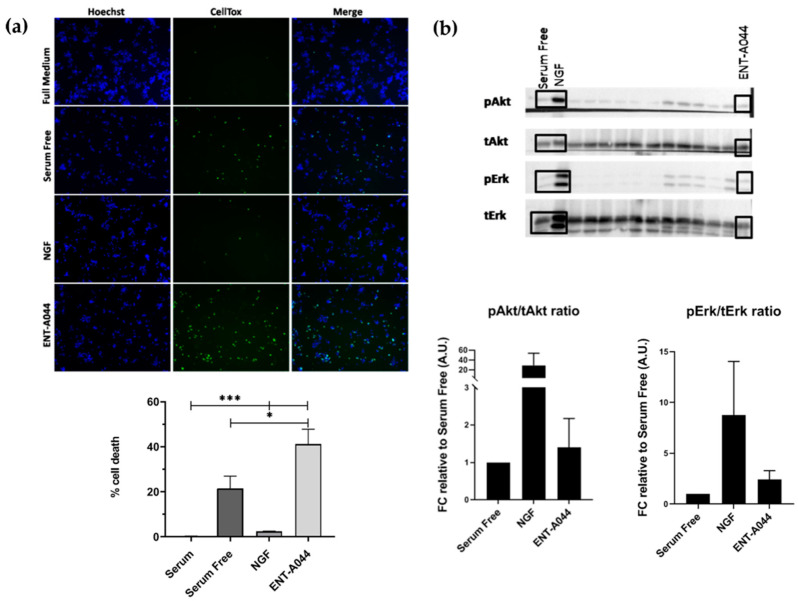
(**a**) Cell tox assay performed on PC12 cells after 24 h and treatment with the tested compound ENT-A044 (500 nM). Quantification of cell tox^+^ cells (green)/Hoechst^+^ cells (blue), one-way ANOVA, * *p* < 0.05, *** *p* < 0.001 mean ± SEM of triplicate measurements. (**b**) Representative blots determined via Western blot analysis of lysates from PC12 cells after 20 min of treatment with NGF (100 ng/mL), synthetic analogs and the compound ENT-A044 (500 nM). Quantification analysis of pAkt and pErk expression (one-way ANOVA against control serum free, mean ± SEM of triplicate measurements).

**Figure 2 ijms-24-11683-f002:**
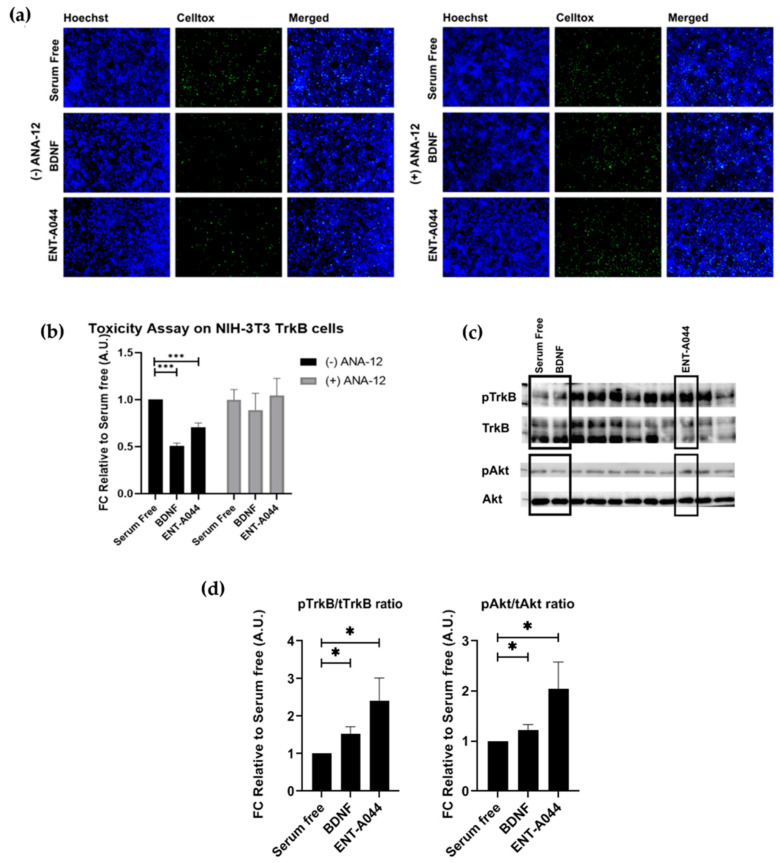
(**a**) Cell tox assay on stable transfected NIH-3T3-TrkB cells after 24 h and treatment with the tested compound ENT-A044 (500 nM). ANA12, i.e., the TrkB inhibitor, was also used. (**b**) Quantification of cell tox^+^ cells (green)/Hoechst^+^ cells (blue), *t*-test against control serum free, *** *p* < 0.001, mean ± SEM of triplicate measurements. (**c**) Representative blots determined via Western blot analysis of lysates belonging to primary mouse cultures of astrocytes after 20 min of treatment with BDNF (500 ng/mL) and the tested compound ENT-A044 (500 nM) (**d**) Quantification analysis of pTrkB and pAkt expression (*t*-test against control serum free, * *p* < 0.05, mean ± SEM of five measurements).

**Figure 3 ijms-24-11683-f003:**
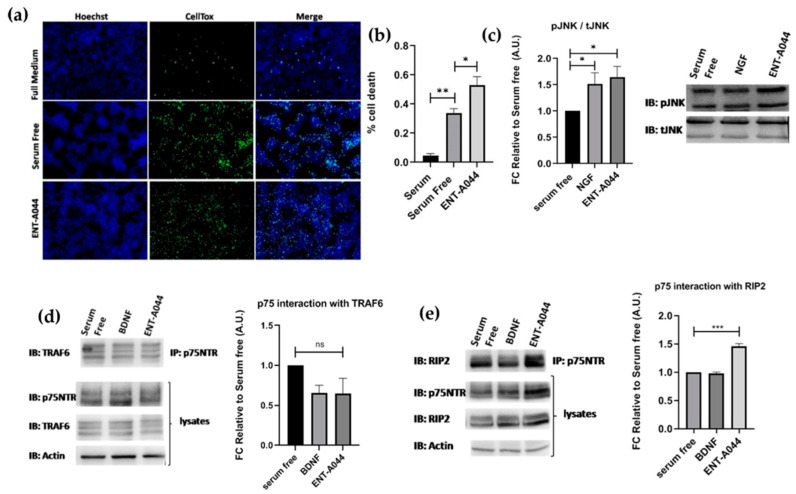
HEK293T cells were transiently co-transfected with the plasmid cDNAs of p75NTR and/or TRAF6 and/or RIP2. (**a**) Cell tox assay performed on transfected HEK293T cells after 48 h and treatment with the tested compound ENT-A044 (500 nM). (**b**) Quantification of cell tox^+^ cells (green)/Hoechst^+^ cells (blue), one-way ANOVA, * *p* < 0.05, ** *p* < 0.01, mean ± SEM of triplicate measurements. (**c**) Western blot analysis performed on transfected HEK293T cells after 30 min of treatment with the tested compound (500 nM) and quantification of pJNK expression (unpaired t-test against control serum free, * *p* < 0.05, mean ± SEM of triplicate measurements). (**d**) Transfectants were exposed for 20 min to BDNF (500 ng/mL), and the tested compound and lysates were immunoprecipitated with p75NTR-specific antibody and immunoblotted with antibodies against TRAF6. Total lysates were analyzed for p75NTR or actin expression via immunoblotting (unpaired t-test against control serum free; ns, no significant; mean ± SEM of triplicate measurements). (**e**) Transfectants were exposed for 20 min to BDNF (500 ng/mL), and the tested compound and lysates were immunoprecipitated with p75NTR-specific antibody and immunoblotted with antibodies against RIP2. Total lysates were analyzed for p75NTR or actin expression via immunoblotting (unpaired t-test against control serum free, *** *p* < 0.005, mean ± SEM of triplicate measurements).

**Figure 4 ijms-24-11683-f004:**
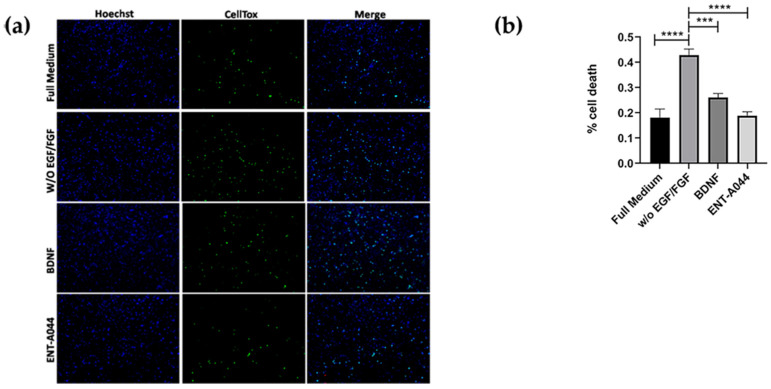
(**a**) Cell tox assay on P7 mouse hippocampal NSCs after 48 h of treatment with the tested compound ENT-A044 (500 nM). (**b**) Quantification of cell tox^+^ cells (green)/Hoechst^+^ cells (blue), one-way ANOVA, *** *p* < 0.005, **** *p* < 0.0001, mean ± SEM of triplicate measurements.

**Figure 5 ijms-24-11683-f005:**
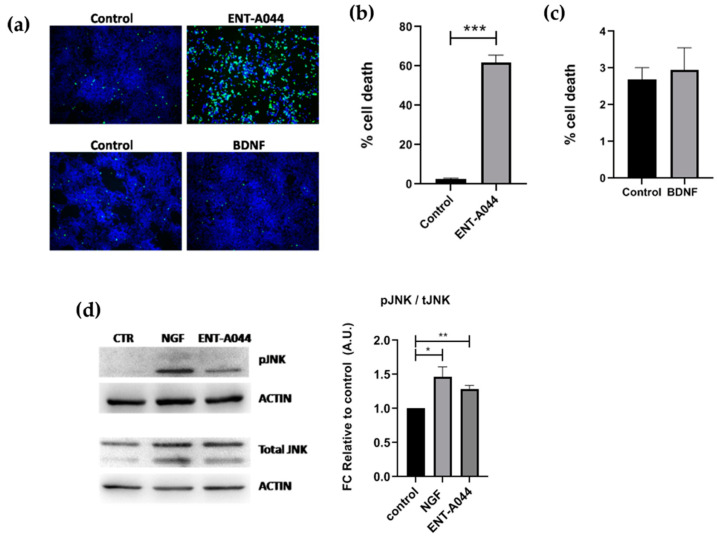
(**a**) Representative images of cell tox assay performed on human NPCs after 48 h of treatment with ENT-A044 (1 μM) and BDNF (500 ng/mL). (**b**,**c**) Quantification of dead cells (green)/Hoechst^+^ cells (blue), *t*-test, ns no significant, *** *p* < 0.001, mean ± SEM of triplicate measurements (**d**) Images and quantification of Western blot analysis of p-JNK and total JNK performed on NPCs treated with ENT-A044 (1 μM) or NGF (100 ng/mL) for 24 h (unpaired *t*-test against control, * *p* < 0.05, ** *p* < 0.005, mean ± SEM of triplicate measurements).

**Figure 6 ijms-24-11683-f006:**
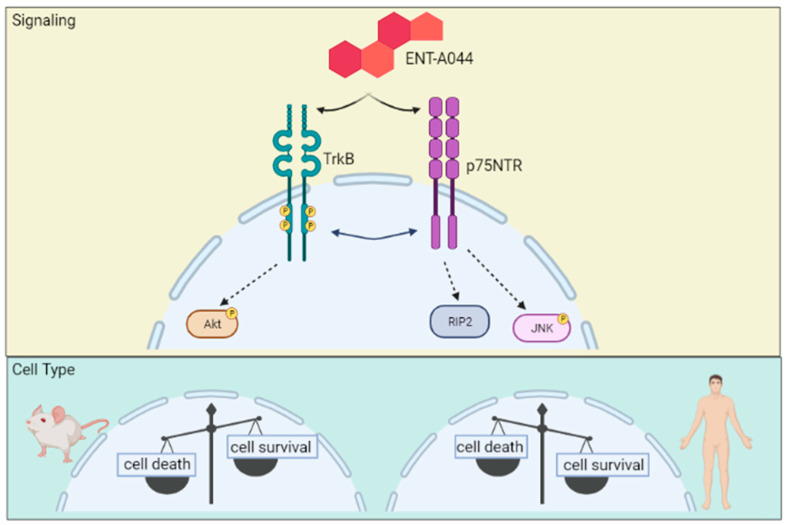
ENT-A044 is shown to result in cell survival in mouse P7 hippocampal NSCs, as it co-expresses both TrkB receptor and p75NTR. In contrast, in iPSCs-derived neuronal precursors, expressing both p75NTR and TrkB receptor, as well as ENT-A044, results in cell death. Thus, ENT-A044 is able to activate both receptors, resulting in differential signaling in a cell type-specific manner, depending on the species of origin—mouse or human—of the neuronal cells (“Figure created with BioRender.com accessed on 12 July 2023”).

## Data Availability

All materials are available upon request. Synthetic compounds are available under a Material Transfer Agreement with the University of Crete, FORTH and the National Hellenic Research Foundation.

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
