# Peer review of "Neurotrophin Analog ENT-A044 Activates the p75 Neurotrophin Receptor, Regulating Neuronal Survival in a Cell Context-Dependent Manner"

_ijms, 2023, doi:10.3390/ijms241411683_

Round 1

Reviewer 1 Report

The manuscript from Papadopoulou et al titled “Neurotrophin analog ENT-A044 activates the p75 Neurotrophin Receptor promoting cell death in mouse and human neuronal cells” addresses the effects of a novel DHEA-derived synthetic analog ENT-A044 in inducing the activation of p75NTR and modulating neuronal cell death and survival in different mouse- and human-derived cell types. The authors show that a treatment with EN-A044 is able to activate both p75NTR and TrkB (but not TrkA). Specifically, EN-A044 induces cell death via the activation of p75NTR, when this is expressed alone. On the other hand, application of EN-A044 is described to result in neuroprotection via TrkB in mouse-derived neuronal cells co-expressing the two receptors. On the contrary in iPSC-derived hippocampal neuronal precursors, expressing both p75NTR and TrkB, application of EN-A044 result in increased cell death. The authors conclude that EN-A044 is able to activate both p75NTR and TrkB resulting in differential signaling and cellular effects in a cell type-specific manner, depending on which receptor is expressed. Moreover, the effects of EN-A044 depend upon the species of origin -mouse or human- of the neuronal cell.

Finding molecules able to cross the blood brain barrier and specifically modulating the different neurotrophin receptors with little or no side effects is of great interest for the therapy of several neurological diseases. This manuscript contributes to increase the knowledge in this context and is therefore very interesting. The experiments are generally well planned and documented and mostly support the conclusions drawn by the authors. I have only a few comments that should be addressed before publication.

1)     I find that the title only partially describes the results presented in this manuscript. I think that it should be changed to better underline the cell type- and species-specific effects;

2)     I would find very useful to have a summary figure accompanying the conclusions/discussion to better visualize the different effects of EN-A044 in the different cellular preparations;

3)     Regarding the presentation of the western blot analysis, I think that the loading control, which is shown in the original pictures, should also be added to all the figures in the manuscript. Also, the partial labelling of the gels (e.g. in figure 1b and 2c) is confusing. Are the non-labelled columns repetitions?

4)     A detailed description of how the western blot were analyzed should be added. Especially, the number of repetitions for each sample in each gel should be given;

5)     Regarding the concentration at which NGF, BDNF and EN-A044 are used: it would be nice to give them all either in ng/ml or in mM. Why is BDNF given to such high concentrations? In the literature concentration between 40 and 100ng/ml are more common. What does this tell about the comparison with EN-A044?

6)     At line 153, data regarding the control experiment with HEK293 cells not expressing p75NTR and treated with EN-A044 are given as “not shown”. I am not sure here what the policy of the journal is, but I think that all data relevant to the interpretation of the results should be shown. Eventually they can be added to the supplementary figures;

7)     Regarding the specificity of the effects of p75NTR upon EN-A044 it would be extremely interesting to also perform the treatment with EN-A044 and simultaneously prevent p75NTR signaling, e.g. using a blocking peptide.   

Minor points:

1)     Correct the spelling mistake at line 96 : “deprovation” instead of “deprivation”

2)     At line 233, revise this phrase “which consist a lipophilic small molecule”

only minor spelling mistakes

Author Response

Rebuttal Letter for manuscript entitled: “Neurotrophin analog ENT-A044 activates the p75 Neurotrophin Receptor, regulating neuronal survival in a cell context dependent manner.” by Papadopoulou et al.

Response to Reviewers’ comments for the article: "Neurotrophin analog ENT-A044 activates the p75 Neurotrophin Receptor promoting cell death in mouse and human neuronal cells."

We would like to thank both reviewers for the time and effort that they dedicated to provide feedback on our manuscript and we are grateful for the insightful comments to our paper. In the present rebuttal letter, we provide all our responses to their suggestions. In addition, we have resubmitted our manuscript highlighting all necessary changes within the manuscript.

Reviewer 1

Comment: I find that the title only partially describes the results presented in this manuscript. I think that it should be changed to better underline the cell type- and species-specific effects.

Reply: Thank you for pointing this out. We totally agree with the reviewer and the new title of our paper is: "Neurotrophin analog ENT-A044 activates the p75 Neurotrophin Receptor, regulating neuronal survival in a cell context dependent manner".

Comment: I would find very useful to have a summary figure accompanying the conclusions/discussion to better visualize the different effects of ENT-A044 in the different cellular preparations.

Reply: We agree that this is a useful suggestion which can improve the understanding of our results. We have made a graphical abstract (Figure 6 on our revised manuscript), where we summarize ENT-A044 ability to activate both receptors, TrkB and p75NTR, resulting in differential signaling in a cell type-specific manner, depending on the species of origin of the neuronal cells.

Comment: Regarding the presentation of the western blot analysis, I think that the loading control, which is shown in the original pictures, should also be added to all the figures in the manuscript. Also, the partial labelling of the gels (e.g. in figure 1b and 2c) is confusing. Are the non-labelled columns repetitions?

Reply: We would like to thank the reviewer for her/his comment. However, we prefer not to add the loading control on the manuscript’s figures in order to present only the necessary results in a more concise way.

According to the labeling of the gels (in figures 1b and 2c): although we agree that this was a lack of proper explanation in the first submission, we now explain that the non-labelled columns are not repetitions. More specifically, these columns represent other synthetic compounds, derived also from DHEA, that were screened along with ENT-A044 for their ability to activate TrkA or TrkB receptor. However, these compounds’ activity will be presented in another manuscript, irrelevant to ENT-A044 effects. That is the reason why we do not provide any information for the non-labelled columns.

Comment: A detailed description of how the western blot were analyzed should be added. Especially, the number of repetitions for each sample in each gel should be given.

Reply: We have now added the suggested content to the manuscript on the section of Material & Methods, where Immunoprecipitation and Immunoblotting were analyzed as suggested.

We would like also to clarify that for the phosphorylated forms of proteins, such as Akt, Erk1/2, TrkB and JNK, the analysis derived from the phosphorylated fraction relative to the total fraction. For the immunoprecipitation assay (activation of p75NTR due to its interaction with the intracellular TRAF6 and RIP2 proteins), the analysis derived from the immunoprecipitated fraction relative to the total fraction. Also, it should be noticed that no repetitions of the same sample are presented in the same gel. Specifically, we depict 3-5 independent experiments where each sample is analyzed with only one repetition.

Comment: Regarding the concentration at which NGF, BDNF and ENT-A044 are used: it would be nice to give them all either in ng/ml or in mM. Why is BDNF given to such high concentrations? In the literature concentration between 40 and 100ng/ml are more common. What does this tell about the comparison with ENT-A044?

Reply: While we appreciate the reviewer’s feedback on measurement units, we think that it is more appropriate to use mM for our synthetic compounds and ng/ml for the endogenous ligands such as NGF and BDNF, since this is the common units that we and other groups have extensively used in previous publications. In this way, the comparisons between similar compounds (steroids or peptidic factors) would be feasible easily.

The reviewer is correct about the concentration of BDNF used in the literature. Upon dose response experiments that we previously performed using different BDNF concentrations, we have observed that the 500ng/ml concentration of this specific BDNF lot is the most effective for activating TrkB receptor. To support this, recent studies from our lab have also used the concentration of 500ng/ml (see Tsimpolis et. al. Biomolecules, 2022; Rogdakis et al. Biomedicines, 2022). 

Concerning the comparison between BDNF and ENT-A044: both molecules can effectively activate TrkB receptor but the underlying mechanism and receptor’s conformational arrangements for each agonist remain unclear. Since, these two molecules present big differences in size, structure and other chemical properties, it is very possible to interact in different regions of the receptor. Such evidence of different interactions of molecules with neurotrophin receptors are recently presented not only from our work (see Pediaditakis et al. Neuropharmacology, 2016 and Frontiers in Pharmacology, 2016) but also from other groups (see Casarotto et al., Cell, 2021).  All these studies introduce a new concept on neurotrophin receptors’ functions in nervous system: cholesterol and its derivatives can interact and modify neurotrophin receptors activity in a different manner -implicating conformational changes, signaling properties, cellular effects- than the endogenous neurotrophic factors.

Comment: At line 153, data regarding the control experiment with HEK293 cells not expressing p75NTR and treated with ENT-A044 are given as “not shown”. I am not sure here what the policy of the journal is, but I think that all data relevant to the interpretation of the results should be shown. Eventually they can be added to the supplementary figures.

Reply: We agree with the reviewer’s assessment and as he/she suggested and we have added all data relevant to the results at Figure 3a, 3b and to the supplementary figure S1. We kept the same figure on our manuscript but in the supplementary data, we also added the sample "untransfected HEK treated with ENT-A044", showing the same levels of cell death as we observed in serum free condition.

Comment: Regarding the specificity of the effects of p75NTR upon ENT-A044 it would be extremely interesting to also perform the treatment with ENT-A044 and simultaneously prevent p75NTR signaling, e.g. using a blocking peptide.

Reply: Although we fully agree that this is an important consideration, we have to admit that after performing many experiments using the p75NTR inhibitor based on MC192 antibody structural properties, which theoretically is blocking receptor’s activity interacting with its extracellular domain (as it is described in several previous studies), we were not able to detect this inhibition in receptor’s activation. Many other reports, as well as our long-term studies, have shown that this potential inhibitor can also act as partial agonist depending on its concentration and cell type. In our hands, we have observed both agonistic and antagonistic results, and thus we are not convinced about the selectivity of its inhibitor’s effects. On the contrary, ANA-12, the selective TrkB inhibitor, was always similarly effective and potent on receptor’s activation, and thus it was used in our study. In order to bypass the lack of chemical inhibitors for p75NTR, we have now designed specific shRNAs against p75 receptor in order to perform genetic elimination of the receptor. We are now testing these shRNAs in different cell types expressing the receptor in order to determine their selectivity and potency.

Reviewer 2 Report

-This is a very interesting manuscript with novel research questions,
detailed experimental methods, and logical and scientific presentation
of results."The study confirms the role of Neurotrophin analog ENT-A044
in the regulation of neuronal cells. It provides a new tool for the
development of relevant new drugs and for the treatment of diseases.
Therefore, I agree to accept the manuscript for publication in its
current form.

-The authors could add their future anticipation for their findings in the end of discussion also should provide the limitations of their study to allow other researchers to continue on this route in future researches.

Author Response

Reviewer 2

Comment: The authors could add their future anticipation for their findings in the end of discussion also should provide the limitations of their study to allow other researchers to continue on this route in future researches.

Reply: We would like to thank the Reviewer for her/his comment. Our discussion section has now been updated to present the limitations of our study as well as its perspectives.
